# Antecedents of Gen Z's green purchase intention in Vietnam's fashion industry with the moderating role of greenwash perception

Anh Duc Do, Dieu Linh Ha ◉*, Thi Thuy Linh Phan, Thi Mai Bui◉,
Tran Bao Ngoc Le, Minh Ngoc Tran◉, Diem Quynh Dang

National Economics University, Hanoi, Vietnam

* linhhd@neu.edu.vn

## Abstract

While buying sustainable fashion items is becoming more and more worldwide, the concept of green fashion is still relatively new in Vietnam. Besides, many fashion brands use "greenwashing" to deceive consumers and encourage their purchases. This study aims to examine the drivers of green purchasing intention among Gen Z in Vietnam's fashion industry with greenwash perception as the moderating factor. The conceptual model was analyzed using structural equation modeling with the bootstrapping method based on data gathered from 467 Vietnamese Gen Z customers. The findings reveal significant positive influences of attitude, subjective norms, perceived behavioral control, and perceived consumer effectiveness on green purchase intention. Greenwash perception serves as a significant negative moderator, enhancing the relationship between the determinants and green purchasing intention. The results hold significant implications for businesses, encouraging them to embrace transparent and genuine sustainability practices instead of engaging in greenwashing. This can be achieved by clearly communicating their initiatives, offering third-party certifications, and conducting educational campaigns. The study provides original contributions to the existing body of literature and offers recommendations for future research in the context of developing countries.

## 1. Introduction

Green consumption has become a global trend, originating in developed countries and gradually spreading to developing nations like Vietnam [1]. To promote environmental conservation, the United Nations introduced the Sustainable Development Goals (SDGs), with Goal 12 emphasizing sustainable production and consumption to reduce the ecological footprint. While researchers have extensively studied green consumption in various fields [2,3], there is still limited research on sustainable fashion consumption, particularly in Asia's emerging markets. Consequently, to broaden

**Data availability statement:** All relevant data are within the article and its supporting information files.

**Funding:** The author(s) received no specific funding for this work.

**Competing interests:** The authors have declared that no competing interests exist.

the body of current research, this study focuses on the Vietnamese fashion sector, where the idea of "green fashion" was still relatively new.

In Vietnam, the government has implemented the National Strategy for Green Growth for 2021–2030 with a vision for 2050. The strategy includes initiatives to encourage sustainable consumption and promote greener lifestyles. Alongside rapid economic growth and a booming fashion sector [4], Vietnamese consumers are increasingly aware of sustainability. However, the sustainable fashion industry in Vietnam faces significant challenges, including high production costs, limited resources [5,6], and deceptive practices like greenwashing, where companies create false impressions of environmental friendliness [7]. Additionally, Vietnam still lacks comprehensive regulations and enforcement mechanisms to address these issues effectively. To be more precise, greenwashing refers to creating a false impression or presenting misleading information about a company's products being environmentally friendly. It arises from combining poor environmental performance with positive communication about sustainability [7]. While greenwashing has been widely studied, there is a notable lack of research on its prevalence in the fashion industry within developing Asian countries [8]. This is crucial, as consumers are often willing to pay more for products from companies genuinely committed to sustainability [9].

The research focuses on Generation Z (born between 1997 and 2012) for their strong awareness of ethical and environmental issues. This generation is often willing to pay more for eco-friendly products and has a significant influence on market trends [10,11]. However, despite their environmentally conscious attitudes, Gen Z remains the largest consumer group in the fast fashion industry, accounting for about 40% of global sales. By 2025, Gen Z will include 2 billion individuals worldwide and will make up approximately 25% of Vietnam's labor force, significantly influencing household and market consumption decisions [12]. Given their dual role as eco-conscious consumers and primary contributors to fast fashion consumption, it is critical to examine Gen Z's green purchase intentions in Vietnam's fashion sector.

Therefore, most studies on green purchasing intention have taken a general approach, with limited focus on the fashion industry [10,13] or specific populations like Gen Z. Furthermore, while greenwashing has been studied as an independent variable [7,14], there is a lack of research exploring its moderating role in shaping consumer behavior. To address these gaps, this study focuses on the Vietnamese fashion industry and investigates the influence of greenwashing perception on Gen Z's intention to purchase green fashion products.

To achieve the research concern, the research will answer the following questions:

*RQ1:* How do four drivers (attitudes, subjective norms, perceived behavioral control, and perceived consumer effectiveness) influence the Gen Z's green purchase intention in Vietnam's fashion industry?

*RQ2*: How does the perception of greenwashing moderate the relationship between these factors and Gen Z's intention to consume green fashion products?

This study provides both theoretical and practical contributions by addressing these questions. By extending the Theory of Planned Behavior (TPB) model with four drivers

(attitudes, subjective norms, perceived behavioral control, and perceived consumer effectiveness) and underscoring the role of moderating variable of greenwashing perception serving, this study provides a thorough understanding of factors influencing green purchasing intention among Gen Z in Vietnam's fashion industry. The research utilized the extension of TPB and used the PLS-SEM method for data analysis. The following section of the study will include literature review and the development of hypotheses. The methodology will be explained in Section 3, and after that, the data analysis techniques and results will be shown in Section 4. The last part provided the conclusions, implications, and study limitations.

## 2. Literature review and hypothesis development

### 2.1. Greenwashing and greenwashing perception

Green marketing has developed in response to consumers' increased desire for products and services that have a positive environmental impact as they become more concerned about environmental protection in the context of globalization [14]. Using green marketing methods has become standard practice for companies looking to get a competitive edge and draw in eco-aware customers. It is imperative to acknowledge that not all assertions made via green marketing precisely mirror an organization's true environmental practices, which may result in "greenwashing" [14]. The phrase "greenwashing" refers to the false and dishonest representations made by certain companies regarding the environmental friendliness of their products or services [15]. It is important to recognize greenwashing because, in the absence of it, well-meaning consumers may be duped into believing that their choices would contribute to environmental preservation. The term "greenwashing" has a lot of definitions, especially in the last several years. The practice of misleadingly disclosing "green" activities is known as "greenwashing" [16,17]. It may be dependent on external factors, incentives, or pressures that define the institutional context in which it takes place [17–20]. Recent years have seen a significant development in the reality of "greenwashing" in the fashion business, particularly as people's awareness of environmental preservation and sustainable living has grown [21,22].

A negative perception component, greenwash perception is defined as the public or consumers recognizing and perceiving inaccurate or deceptive comments made by businesses about their environmental actions [23]. This view reflects people's misgivings about whether a corporation is carrying out its environmental protection obligations and their cynicism about the sincerity of the company's green marketing initiatives [24]. Customers generally hold the view that environmentally friendly products are safer and more ecologically friendly [25], which results in a generally favorable perception of these items [26]. Customers are more inclined to doubt advertising claims when they see examples of greenwashing, though, and this can have a detrimental effect on how they feel about the products [27]. Thus, greenwash perception has emerged as a key metric for assessing consumer trust and business social responsibility initiatives in the present climate of growing environmental consciousness [21].

### 2.2. Green purchase intention

Green purchase intention refers to the consumer's willingness or readiness to purchase environmentally friendly products [28]. When evaluating customers' green intentions, several factors are crucial, including beliefs, needs, values, motivation, knowledge, demography, and attitudes [29]. A person's inclination and desire to favor environmentally friendly products over non-eco-friendly ones is theorized while making a purchase decision [30]. Customers identify their need for a product during the evaluation process, which influences their choice to buy [31]. In the context of green purchasing, this intention manifests when consumers consciously decide to buy eco-friendly products, aligning their actions with a commitment to reducing environmental harm [32]. Based on the TPB model, green purchase intention builds on the broader concept of general purchase intention by raising consumer awareness of environmental sustainability, health, and nature while concentrating on eco-friendly products [33]. Research suggests that a positive attitude toward green products significantly enhances the likelihood of forming green purchase intention [34]. Green purchase intention varies across cultures,

individuals, and genders, highlighting the contextual nature of consumer behavior [35]. Identifying and appreciating green attributes in products not only motivates consumers to make environmentally conscious decisions but also positions them as responsible citizens who contribute to broader societal changes [36,37].

## 2.3.  Extended Theory of Planned Behaviour (TPB)

The TPB of Ajzen [38,39] is a decision-making model that predicts behavioral intention through three components: Attitude, Perceived Behavioral Control, and Subjective Norms. See Fig 1 shows the TPB framework. Moreover, Ajzen [39] highlights the model's flexibility, allowing the addition of new variables to enhance its predictive power. For the sake of this research, perceived consumer effectiveness – the belief in one's ability to make a positive environmental impact – is incorporated as an additional factor influencing intentions [40,41]. Therefore, this paper applies the extended TPB as a framework for understanding purchasing green product choices. Extended TPB has demonstrated its ability to explain and predict both ethical and unethical conduct in many aspects of life, e.g., health, cheating, recycling, and green purchase [3,42–44] and is one of the most frequent theories used to investigate energy-saving behavior [45].

In the context of fashion and sustainable fashion, the extension of TPB has proven to be a useful model for forecasting consumer intentions, particularly Gen Z consumers. [46] used TPB extensions framework analysis to assess the reasons behind purchasing luxury fashion clothes manufactured in sweatshops. [46] endeavored to verify extended TPB framework in the context of green product consumption, which encompasses environmental concerns. The authors discovered that while customers' concerns about greenwashing have a detrimental impact on this relationship, attitude toward sustainable clothes influences purchase intention most. A model based on the extension of TPB was developed in Ge's [47] research to determine the elements of influencer marketing that impact the primary reasons why Gen Z consumers intend to buy sustainable fashion products. Therefore, this study leverages the extension of the TPB model by adding perceived consumer effectiveness, with its adaptability and strong track record, to explore the drivers of green consumer behavior in the context of sustainable fashion.

## 2.4.  Hypothesis development

### 2.4.1.  Attitudes.  One of the most significant factors influencing consumers' shopping decisions is their attitude [39,48]. The degree to which a person assesses or communicates favorability or unfavorability toward the action in issue is known as their attitude toward behavior [39]. A person's attitude has an intentional impact on their intention to purchase items because those with strong attitudes appear to be more driven to do so [49]. When making purchases, environmentally conscious customers are also more concerned about good quality, safety, and health., and the requirements of other people [2]. A variety of research evidence shows that attitudes and the intentions and behaviors of buying environmentally friendly products are closely related [50]. If the attitude is more positive toward a specific behavior, the likelihood of executing that behavior is also higher. According to [51], the attitudes of consumers are acknowledged as a crucial determinant in attaining green consumption behavioral intention. Numerous studies have demonstrated that, across

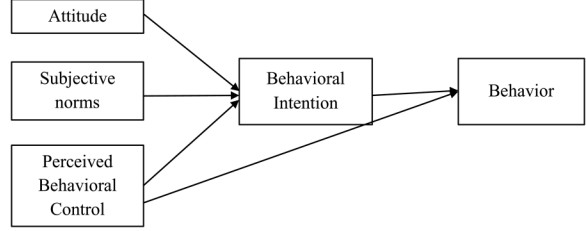

**Fig 1.  Theory of planned behavior.** Fig 1 shows the TPB framework.

cultures, countries, and product categories, there is a positive correlation between consumers' views and their intentions to make green purchases [52,53].

Customers who are pro-environmental protection issues typically show a preference for buying more sustainable fashion items. Hence, we propose the below hypothesis:

**H1: Attitude toward green products positively influences green purchasing intention.**

**2.4.2. Subjective norms.** Subjective norms are the expectations of an important group or an individual that a particular behavior will be accepted and encouraged. Subjective norms are shaped by the social pressure that people perceive from others to act in a particular way, as well as the incentive that compels people to adopt the opinions of other people [54]. Additionally, according to [55], "A sort of opinion that people approve or disapprove about a certain behavior which is undertaken and performed" is how it is defined.

There exists a positive association between green consumption and ethical ideals. Subjective norms, such as ecologically responsible shopping, can more successfully influence an individual's environmental intention because environmental activity, particularly in the field of environmental protection, has moral appeal [56]. The willingness to utilize green products is influenced by ethical commitments and environmental ethics [57]. Because customers believe that green products are more ecologically friendly than other products, they may choose to support or oppose them in response to societal pressure [28]. Accordingly, in sustainability research on human decision-making that considers the social environment, subjective standards are seen as a major determinant. Since environmentally harmful actions indirectly affect other people, it is possible to avoid them in social situations or replace them with actions that are better for the environment [46]. According to [58], there was a substantial correlation between the subjective norm and the intended as well as actual decision preferences. Thus, we believe that under the subjective norm, consumers will have strong intentions to make green purchases. Many consumers perceive the use of green products as ethical behavior and are influenced by societal attitudes toward environmental protection. This leads to the hypothesis:

**H2: Subjective norms positively influences green purchasing intention.**

**2.4.3. Perceived behavioral control.** Perceived behavioral control is the product of self-efficacy (perceived capacity to do a task) and control belief (belief in the level of control one has over events and outcomes in one's life). It includes situational control awareness and awareness of one's capabilities [59]. TPB emphasizes how important a person's intrinsic personality and character are when making judgments. TPB therefore supports considering psychological factors while making self-regulated and socially influenced decisions [60]. Perceived behavioral control can be used to predict young consumers' desire to buy environmentally friendly products [13]. Furthermore, a UK study by [61] found that customers' perceived behavioral control is determined by the availability of green items, which affects their propensity to buy. Therefore, the hypothesis is proposed:

**H3: Perceived behavioral control positively influences green purchasing intention.**

**2.4.4. Perceived consumer effectiveness.** Perceived consumer effectiveness is conceptualized as an individual's belief that their efforts in action can make a difference in addressing environmental issues [40]. Intention and actual conduct are both dependent on the degree of perceived consumer effectiveness, which arises when customers believe their activities have an impact [62]. Socially conscious customers prefer to take into account the social impact of their purchases when making green purchases because they feel empowered to contribute to the reduction of pollution [63]. Jaiswal & Kant [64] together with Straughan & Roberts [65] have verified the significance of perceived consumer effectiveness as a precursor to both green buying intentions and actions. Since perceived consumer effectiveness shows a stronger correlation with green buying intention than environmental attitude does, It is a more accurate predictor of green purchasing behavior than just attitude toward the environment.

In the context of electromobility [66], discovered that perceived consumer effectiveness positively moderates the association between the formation of consumer attitudes and intentions to use environmentally friendly products. Additionally, it was discovered that perceived consumer effectiveness significantly moderated the association between green practices and green attitudes toward businesses that use sustainable operations [67]. Therefore, the hypothesis is proposed:

***H4: Perceived consumer effectiveness positively influences green purchasing intention.***

**2.4.5. Moderating role of greenwash perception.** Consumers with positive attitudes toward green products are more likely to exhibit green purchase behavior [3,68]. However, greenwash perception can moderate this relationship. If consumers perceive greenwashing, their positive attitude towards green products may weaken, leading to a decreased likelihood of green purchases.

Subjective norms represent societal influences on consumer behavior [69]. Consumers are affected by their social circles and the perceived social acceptability of various activities. Greenwash perception might help to attenuate this association. If consumers learn of a company's greenwashing, they may be less likely to buy the product because of fear of social repercussions for supporting ecologically hazardous actions.

Perceived behavioral control relates to customers' belief in their capacity to do a given activity [69]. In the context of green purchase intention, this translates to a consumer's confidence in their ability to make green choices. Studies have shown a positive correlation, indicating that consumers who feel empowered to make a difference through their purchases are more likely to do so. However, when consumers encounter greenwashing tactics, it can erode their trust in companies' environmental claims. This skepticism can lead them to question the effectiveness of their individual choices, hence, discouraging consumers from engaging in green purchasing despite their perceived ability to do so.

Similar to perceived behavioral control, perceived consumer effectiveness plays a role in influencing purchasing intention and refers to a consumer's belief in the impact of their actions on achieving a desired outcome. In the context of green practices, this translates to a consumer's belief that their green purchases can make a difference in the environment. However, greenwash perception may undermine this relationship. Exposed to greenwashing, consumers may doubt the true environmental benefit of green products, leading them to question the effectiveness of their choices despite a strong belief in making a difference (see Fig 2).

Therefore, we propose the following hypotheses.

***H5: Greenwash perception negatively moderates the relationship between attitude and green purchasing intention.***

***H6: Greenwash perception negatively moderates the relationship between subjective norms and green purchasing intention.***

***H7: Greenwash perception negatively moderates the relationship between perceived behavioral control and green purchasing intention.***

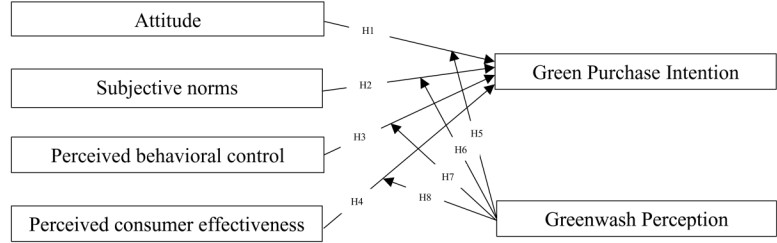

**Fig 2. Hypothesized model.** The paper proposes 8 hypotheses.

*H8: Greenwash perception negatively moderates the relationship between perceived consumer effectiveness and green purchasing intention.*

## 3. Methodology

### 3.1. Data collection and sample

The authors collected data through a quota convenience sampling method. The survey targeted in this study were Vietnamese Generation Z customers, who were born between 1997 and 2012. Data collection was conducted anonymously through an online survey distributed via Google Forms across various media channels and forums. This approach helped to maintain participant neutrality and enhance data accuracy. A total of 500 responses were recorded, of which 467 were deemed valid after excluding incomplete or inconsistent entries. The survey achieved a high completion rate of 93%, reflecting strong participant engagement.

Out of 467 appropriate respondents, there are 28.9% of male respondents compared to 71.1% female respondents. Of the Gen Z participants surveyed, 76.9% are 18–22 years old, 17.3% are between the ages of 13 and 17, and 5.8% are between the ages of 23 and 28. Most survey respondents reside in Northern Vietnam, with 343 individuals (73.4%), followed by Southern Vietnam with 66 respondents (14.1%), and Central Vietnam accounting for 9.4%. Regarding occupation, the largest group comprises students, totaling 434 participants (92.9%), followed by employees with 23 individuals (4.9%). Business owners, freelancers, and others make up 0.9%, 0.4%, and 0.9%, respectively. Most participants spend less than 500,000 VND per month on green products (84.2%), while a smaller proportion spend between 500,000 and 1,000,000 VND (14.1%), and only a very small percentage spends over 1,000,000 VND (1.7%). The descriptive statistics of the survey are shown in Table 1 below.

### 3.2. Research measures

Multiple items on a 5-point Likert-type Scale (1 = "Strongly disagree" to 5 = "Strongly agree") will be used to measure each concept. Every item featured a distinct keyword that embodied the specific concept. The items were grounded in previously validated scales and existing research, clearly indicating the concept they were measuring.

Participants' attitudes toward green purchasing intentions use [70] and [71] measures, as these scales are comprehensive and have been validated in prior research on environmental behavior. Subjective norms were measured using scales from [70], which are widely recognized for their reliability in capturing social influences on behavior. Items related to perceived behavioral control were taken from [70] study, as their measures effectively address individuals' confidence in performing green behaviors. Measures for perceived consumer effectiveness originate from a scale developed by [72,73], chosen for their robust ability to gauge consumers' belief in their impact on environmental outcomes. For evaluating green purchasing intention, the authors use items established by [70]. Finally, greenwashing perception was principally examined using measures developed by [23] and [46], selected for their strong theoretical underpinnings and proven effectiveness in analyzing skepticism toward corporate environmental claims. To confirm the validity and clarity of the assessment measures, the authors first conducted a pilot testing phase that was utilized to be carried out with experienced participants before the formal data collection with 20 Vietnamese Gen Z customers. Minor changes were made to the questionnaire based on the comments received. This pretest will examine the completeness, clarity, language, structure, and appropriateness of the assessment items.

### 3.3. Data analysis

The proposed research model includes four variables from the TPB extended model and the moderated effect of perceptions of greenwashing, that affect consumers' intentions to make green purchases. The authors utilized the Partial Least Squares Structural Equation Modeling (PLS-SEM) method since it requires latent components and various indicators

**Table 1. Descriptive statistics of participants' demographic.**

| Dimensions | Frequencies | % |
|---|---|---|
| *Age (years)* | | |
| 13 - 17 | 81 | 17.3% |
| 18 - 22 | 359 | 76.9% |
| 23 - 28 | 27 | 5.8% |
| *Gender* | | |
| Male | 135 | 28.9% |
| Female | 332 | 71.1% |
| *Residence* | | |
| Northern Vietnam | 343 | 73.4% |
| Central of Vietnam | 44 | 9.4% |
| Southern Vietnam | 66 | 14.1% |
| *Occupation* | | |
| Student | 434 | 92.9% |
| Employee | 23 | 4.9% |
| Business | 4 | 0.9% |
| Freelancer | 2 | 0.4% |
| Others | 4 | 0.9% |
| *Spending on green products monthly* | | |
| Below 500.000 VND | 393 | 84.2% |
| 500.000 VND – 1.000.000 VND | 66 | 14.1% |
| Above 1.000.000 VND | 8 | 1.7% |

Source: Author's compilation, 2024.

to quantify perception [74]. PLS-SEM, which is based on the theory of structural equation modeling, has the following benefits: (1) PLS-SEM employs two independent models, the measurement model and the structural model, to effectively combine latent and observable variables into a single analytical framework [75]. (2) It allows for causal-predictive analysis of complex data with minimum theoretical backing, avoiding the necessity for precise data distribution assumptions [75]. (3) PLS-SEM shows better resistance to variation problems brought on by measurement mistakes than regression analysis, producing more accurate and dependable results [76,77]. The collected data was processed following the procedures outlined by [75], which included assessing the measurement model to ensure reliability and validity, evaluating the structural model to analyze relationships between constructs, and examining the moderating effects of greenwashing perception to understand its influence on green purchasing intentions. Lastly, the authors also used 5000 bootstrapped samples to investigate the modėating effects within the proposed model.

## 4. Results

### 4.1. Hypothesis testing

Results were obtained using Partial least square structural equation modeling (PLS-SEM) version 4.0. Several tests mainly related to reliability, validity, and path coefficients, confirm that the measured data do not have multicollinearity and other data-related deviations [78]. This analysis section used a two-way approach to assess the results: The Measurement model and the Structural model.

 **4.1.1. Measurement model.** The first step in the assessment of the reflective measurement model involves examining the loading indicators. The indicator reliability should be higher than 0.70 [79,80]. Fig 3 shows the value of outer loading,

there were three values which did not meet the requirement (GW1 = 0.628, GW2 = 0.519, PCE1 = 0.650), therefore these scales will be excluded before the second step is conducted) (See Fig 3).

Fig 4 indicates the loading indicator after removing GW1, GW2, and PCE1. All the loading indicators are above 0.7, which can be considered a positive sign of the reliability and validity of the research model, as it indicates that the observed variables have a strong connection with the latent factors (See Fig 4).

The next step is to assess the reliability of internal consistency. To evaluate the construct measures' internal consistency reliability, the Composite Reliability (CR) is considered [81]. Table 3 reflects the CR coefficient values of the hypotheses, which are in the range of 0.865 to 0.932. The rule of thumb used for the Composite Reliability value is higher than 0.7 so the value of Composite Reliability has been fulfilled because it has a value above 0.7. It can be concluded that the construct has good reliability, or the scale used as a tool in this study is reliable or consistent.

As [82] suggest, the individual reliability of the individual items also should be considered by using Cronbach's alpha values as a measure for the homogeneity of a construct, where the value of 0.7 was considered acceptable [83]. The CR should be higher than 0.70 in exploratory research, but a value of 0.60 to 0.70 was also considered acceptable [79]. [75] stated that the minimum value is 0.70 (or 0.60 in exploratory research). Table 2 shows the measurement of Cronbach's Alpha (CA), which varies from 0.770 to 0.912, shows that this study adequately meets the standard for reliability of the measures.

To estimate convergent validity [84], propose calculating the Average Variance Extracted (AVE) by the reliability of the component score for the latent variable and the outcome is more conservative than the composite reliability. The AVE value should be more than 0.50, which indicates that 50% or more of the variance from the indicators can be explained [85]. The value AVE of this study; therefore; is acceptable for the assessment of validity (its minimum value is 0.572).

To assess the existence of discriminant validity [86], proposed a heterotrait-monotrait ratio (HTMT) correlation [87]. The discriminant validity problem is present when the HTMT value is high. [86] proposed a threshold value of 0.90. Table 3 illustrates the values of the HTMT ratio, which are below 0.9; therefore, meet the requirements.

**4.1.2. Structural model.** This article utilized PLS bootstrapping with 5000 bootstraps and 467 cases to enlighten the path coefficients and their significance. The PLS structural model and hypotheses are assessed by examining the significance of the path coefficients and the variance accounted for by the antecedent constructs.

Inner VIF was considered to ensure multicollinearity and data-related deviations. [88] selected a multicollinearity tolerance level of 30% as the cut-off criterion based on guidelines suggesting tolerance values greater than 35% may cause multicollinearity problems. Based on the tolerance levels reported by [88,89] recommend a 33% tolerance level, translating to a VIF of ≤ 3.3. Table 4 below demonstrates the inner VIF, in which all values are smaller than 3.3. Hence there is no issue of multicollinearity.

[90] recommend $R^2$ that endogenous variables be ≥ 0.10. An $R^2$ ≥ 0.10 ensures that the variance explained by the endogenous variables has practical, as well as statistical, significance. In this study, the value of the R-square which is 0.498 is satisfactory.

Table 5 demonstrates that the results of the structural model and the standardized path coefficient indicated positive effects among the constructs in the structural model.

The results show that attitude has a positive effect on consumers' green purchase intention (H1: b = 0.122, t = 3.141, p = 0.002) specifying that H1 was supported. This underscores the role of positive perceptions of sustainable fashion in shaping Vietnamese Gen Z consumers' intentions, aligning with previous research [3,7]. Gen Z's recognition of sustainable fashion as environmentally beneficial and as a representation of their self-image enhances their likelihood of engaging in green purchases. Subjective norms significantly affect Gen Z consumers' green purchase intention (H2: b = 0.234, t = 5.810, p = 0.000), supporting H2. This suggests that societal trends and peer encouragement motivate Gen Z consumers in Vietnam to adopt sustainable purchasing habits, aligning with prior studies [46,58]. Perceived behavioral control positively affects young consumers' green purchase intention (H3: b b = 0.262, t = 6.156, p = 0.000). Therefore, H3

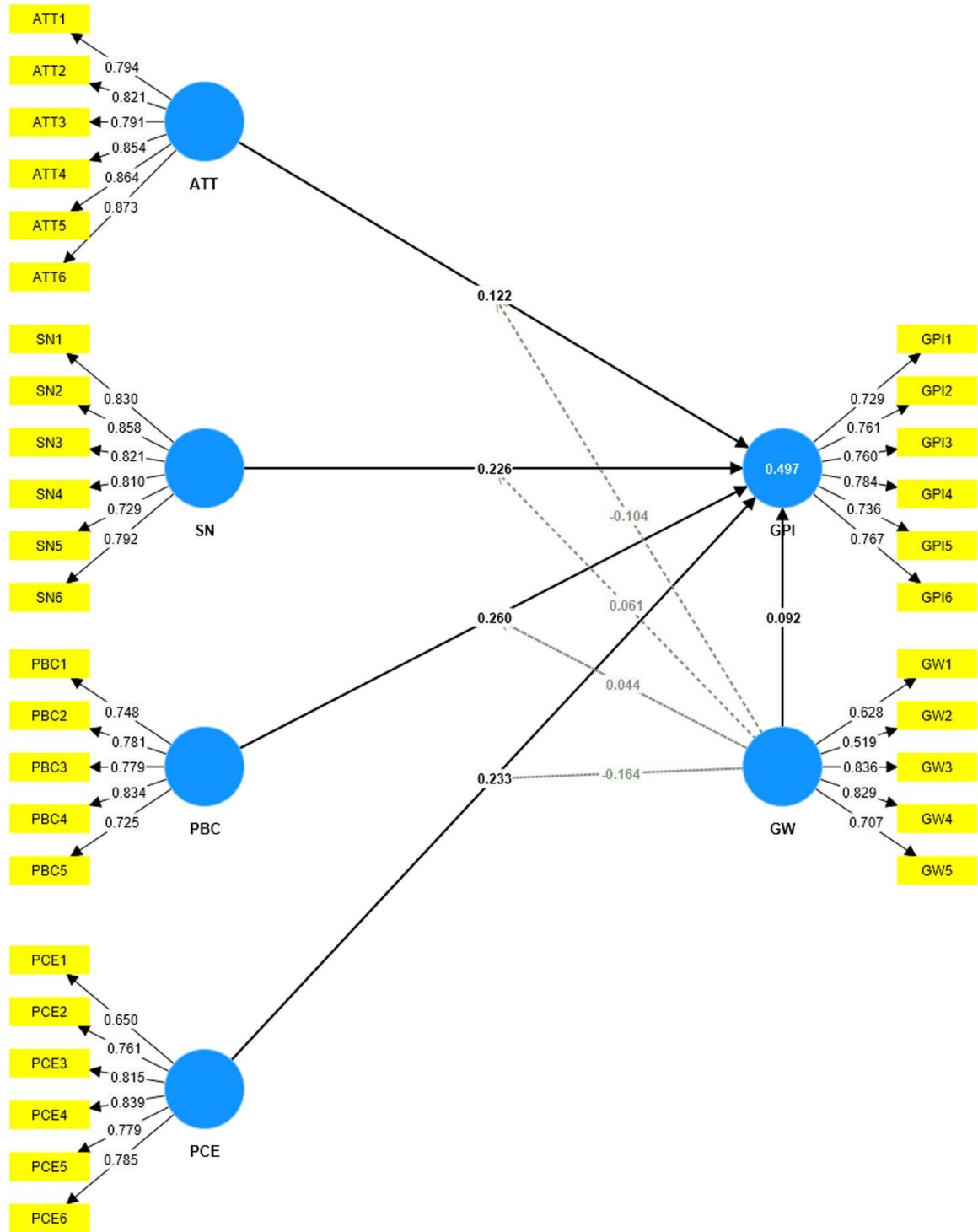

**Fig 3. Initial outer loading.** It shows the value of outer loading, there were three values which did not meet the requirement.

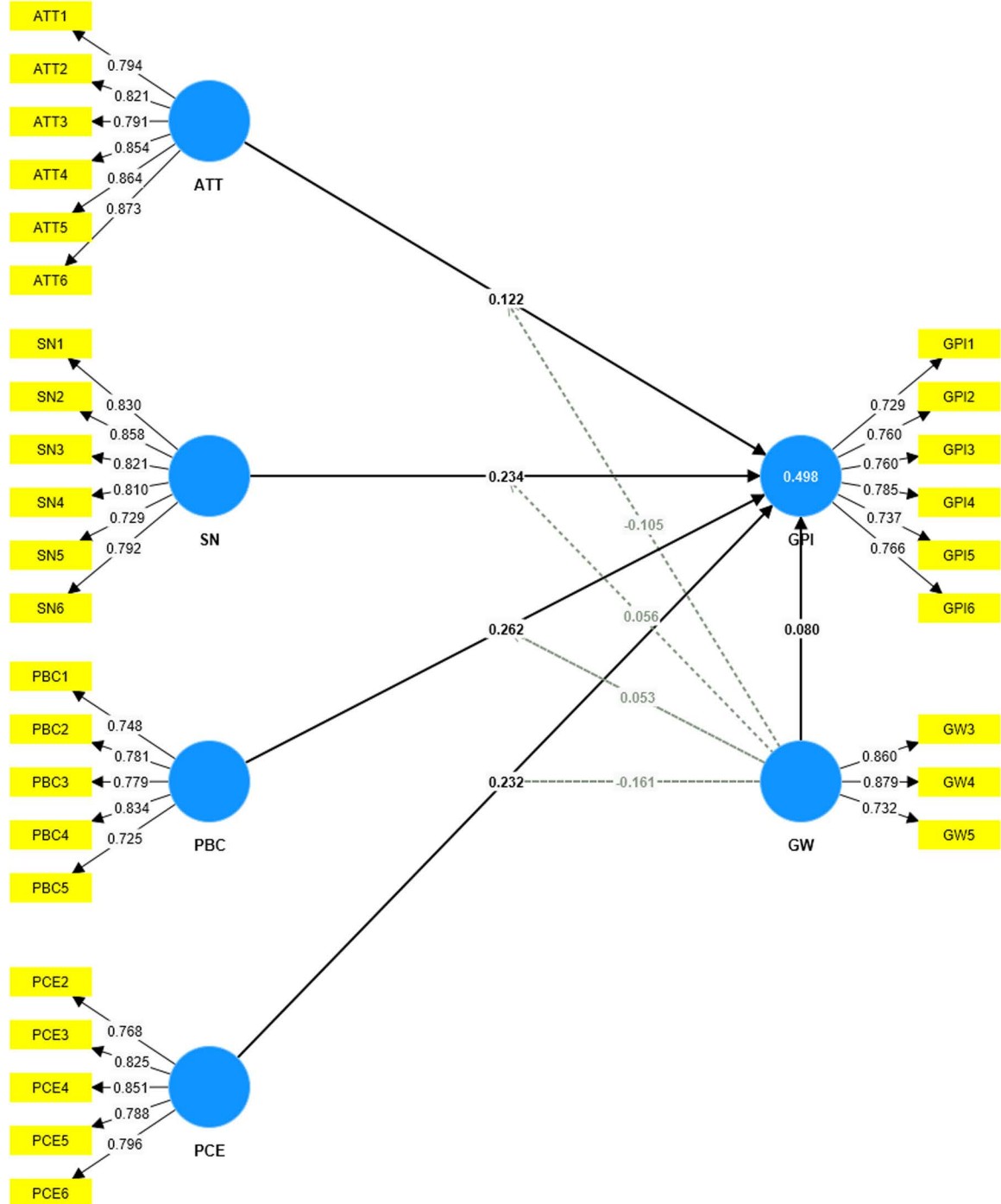

**Fig 4. The outer loading after removing PCE1, GW1, GW2.** Fig 4 indicates the loading indicator after removing GW1, GW2, and PCE1.

is supported. This indicates that confidence and access to sustainable fashion products enhance Gen Z consumers' likelihood of engaging in green consumption, consistent with the results of [60] and [13]. Finally, perceived consumer effectiveness positively affects young consumers' green purchase intention (H4: b = 0.232, t = 5.863, p = 0.000). The findings

**Table 2. Construct reliability and validity.**

| ATT | CA | CR (a) | CR (c) | AVE |
| --- | --- | --- | --- | --- |
| ATT | 0.912 | 0.919 | 0.932 | 0.695 |
| GPI | 0.851 | 0.854 | 0.889 | 0.572 |
| GW | 0.770 | 0.807 | 0.865 | 0.683 |
| PBC | 0.833 | 0.842 | 0.882 | 0.600 |
| PCE | 0.865 | 0.866 | 0.903 | 0.650 |
| SN | 0.893 | 0.896 | 0.918 | 0.652 |

Source: Author's compilation, 2024.

**Table 3. Heterotrait-monotrait ratio (HTMT).**

| | Heterotrait-monotrait ratio (HTMT) |
| --- | --- |
| GPI↔ATT | 0.414 |
| GW↔ATT | 0.334 |
| GW↔GPI | 0.407 |
| PBC↔ATT | 0.307 |
| PBC↔GPI | 0.589 |
| PBC↔GW | 0.371 |
| PCE↔ATT | 0.321 |
| PCE↔GPI | 0.576 |
| PCE↔GW | 0.386 |
| PCE↔PBC | 0.404 |
| SN↔ATT | 0.262 |
| SN↔GPI | 0.490 |
| SN↔GW | 0.274 |
| SN↔PBC | 0.292 |
| SN↔PCE | 0.314 |

Source: Author's compilation, 2024.

suggest that Gen Z consumers are more inclined to make green purchases when they believe their actions contribute to environmental sustainability, reinforcing prior studies by [64] and [71].

The effective size can be used to estimate the power of the moderating effect [82,91]. Based on the $f^2$ value, the effect size of the omitted construct for a particular endogenous construct can be determined such that 0.02, 0.15, and 0.35 represent small, medium, and large effects, respectively [91].

The power of the moderating effect is shown in the Table 6. According to [92], a small effect size does not always imply that the causal preservative effect is insignificant. [92] stated that even a small interaction effect could make sense under exceptional moderating settings; if the resulting beta changes are expressive, at that time, it is important to consider these conditions. This recommended that the moderating effects of greenwash perception in attitude, and perceived consumer effectiveness on the green purchase intention of Gen Z consumers could be significant.

Table 7 below demonstrates the value of $Q^2$. According to [75], the level of $Q^2$ corresponds to the model's predictive ability as follows: $0 < Q^2 < 0.25$: low level of authentication accuracy; $0.25 \leq Q^2 \leq 0.5$: Average forecast report accuracy; $Q^2 > 0.5$: High forecast accuracy.

The value of $Q^2$ in the table indicates the medium predictive relevance.

**Table 4. Collinearity statistics (VIF) – Inner model.**

|  | VIF |
|---|---|
| ATT→GPI | 1.283 |
| GW→GPI | 1.287 |
| PBC→GPI | 1.266 |
| PCE→GPI | 1.353 |
| SN→GPI | 1.327 |
| GW*ATT→GPI | 1.267 |
| GW*SN→GPI | 1.200 |
| GW*PBC→GPI | 1.310 |
| GW*PCE→GPI | 1.209 |

Source: Author's compilation, 2024.

**Table 5. Path coefficients and hypotheses testing.**

| Hypothesis | Relationships | Beta | Mean | SD | t value | p-value | Decision |
|---|---|---|---|---|---|---|---|
| **Direct effect** | | | | | | | |
| H1 | ATT→GPI | 0.122 | 0.127 | 0.039 | 3.141 | 0.002 | Supported |
| H2 | SN→GPI | 0.234 | 0.236 | 0.040 | 5.812 | 0.000 | Supported |
| H3 | PBC→GPI | 0.262 | 0.259 | 0.043 | 6.156 | 0.000 | Supported |
| H4 | PCE→GPI | 0.232 | 0.234 | 0.040 | 5.863 | 0.000 | Supported |
| **Moderator effect** | | | | | | | |
| H5 | GW*ATT→GPI | −0.105 | −0.103 | 0.033 | 3.185 | 0.001 | Supported |
| H6 | GW*SN→GPI | 0.056 | 0.059 | 0.038 | 1.488 | 0.137 | Not supported |
| H7 | GW*PBC→GPI | 0.053 | 0.051 | 0.041 | 1.282 | 0.200 | Not supported |
| H8 | GW*PCE→GPI | −0.161 | −0.157 | 0.033 | 4.848 | 0.000 | Supported |

Source: Author's compilation, 2024.

**Table 6. Strength of moderating effects based on the rule of Cohen 1998.**

| Endogenous latent variable | $R^2$ | Interaction terms | $f^2$ | Effect size |
|---|---|---|---|---|
| Green purchase intention | 0.498 | GW*ATT | 0.021 | Small |
| | | GW*SN | 0.005 | No effect |
| | | GW*PBC | 0.006 | No effect |
| | | GW*PCE | 0.051 | Small |

Source: Author's compilation, 2024.

## 4.2. Moderating effects testing

To model the moderating effects of latent variables in structural equation models [93], proposed building product terms between the indicators of the latent independent variable and the indicators of the latent moderator variable. These product terms serve as indicators of the interaction term in the structural model. In the present research, the product indicator technique is used to identify and evaluate the moderating power of the effect of greenwash perception on the association (ATT-GPI, SN-GPI, PBC-GPI, PCE-GPI).

**Table 7. LV prediction summary.**

|  | $Q^2$predict | RMSE | MAE |
|---|---|---|---|
| GPI | 0.47 | 0.731 | 0.575 |

Source: Author's compilation, 2024.

Table 5 and Fig 5 suggest that the GW*ATT interaction terms (b = − 0.105, t = 3.185, p = 0.001) and the GW*PCE inter-action term (b = −0.161, t = 4.848, p = 0.000) were significant (See Fig 5). Therefore, H5 and H8 have received full support. This suggests that when consumers perceive high levels of greenwashing, their positive attitudes toward sustainable fashion and belief in personal efficacy are diminished, leading to lower purchase intention. These findings align with [46] and [94], emphasizing the significant impact of greenwash perception as a barrier to green consumption. However, the GW*SN interaction terms (b = 0.056, t = 1.488, p = 0.137) and the GW*PBC interaction term (b = −0,053, t = 1.282, p = 0.200) were insignificant. Thus, H6 and H7 have not received support, showing that social influences and perceived control over green purchases remain influential despite skepticism toward corporate sustainability claims.

The slope for the association between attitude, perceived consumer effectiveness, and consumers' green purchase intention moderated by perceived greenwash was given in Figs 6 and 7 respectively (See Figs 6 and 7).

## 5. Conclusions and implications

### 5.1. Conclusions

Building upon the extended TPB framework by incorporating perceived consumer effectiveness as an additional determi-nant of purchase intention, this study examines the factors affecting the green purchase intentions of Gen Z consumers in Vietnam's fashion sector under the moderating role of greenwash perception.

The findings confirm that attitudes, subjective norms, perceived behavioral control, and perceived consumer effective-ness all significantly and positively influence green purchase intention (H1–H4). Among these, perceived behavioral con-trol and perceived consumer effectiveness have the strongest positive impact, highlighting the significance of consumer confidence and belief in personal effectiveness in driving sustainable purchases. While attitude and subjective norms also positively influence green purchase intention, their effects are comparatively weaker. Furthermore, the results of this study also address the limitations of previous studies by demonstrating an inverse moderate influence of greenwash percep-tion on the connection between green purchasing intention and its antecedents. The results demonstrate that greenwash perception weakens the relationships between attitude, perceived consumer effectiveness, and green purchase intention, suggesting that skepticism toward corporate green claims reduces the impact of positive attitudes and perceived effective-ness on green purchase decisions. However, greenwash perception does not significantly moderate the effects of subjec-tive norms and perceived behavioral control, indicating that social influence and accessibility remain strong motivators for green consumption despite greenwashing concerns. This research emphasizes the significance of greenwash perception as a moderating factor and extends the TPB framework to better understand green purchase intentions in Vietnam's fash-ion industry.

### 5.2. Theoretical implications

The research has developed an extended TPB model to study the factors influencing green purchase intention, with the addition of the moderating effect of greenwashing perception. Focusing on generation Z customer in Vietnam's fashion sector, the study introduces a new and different theoretical dimension from previous research by adding the moderating role of greenwash perception has introduced a new dimension to the TPB framework, suggesting that external factors can influence the relationship between the antecedents of green purchase intention. Our results align with [46] argument

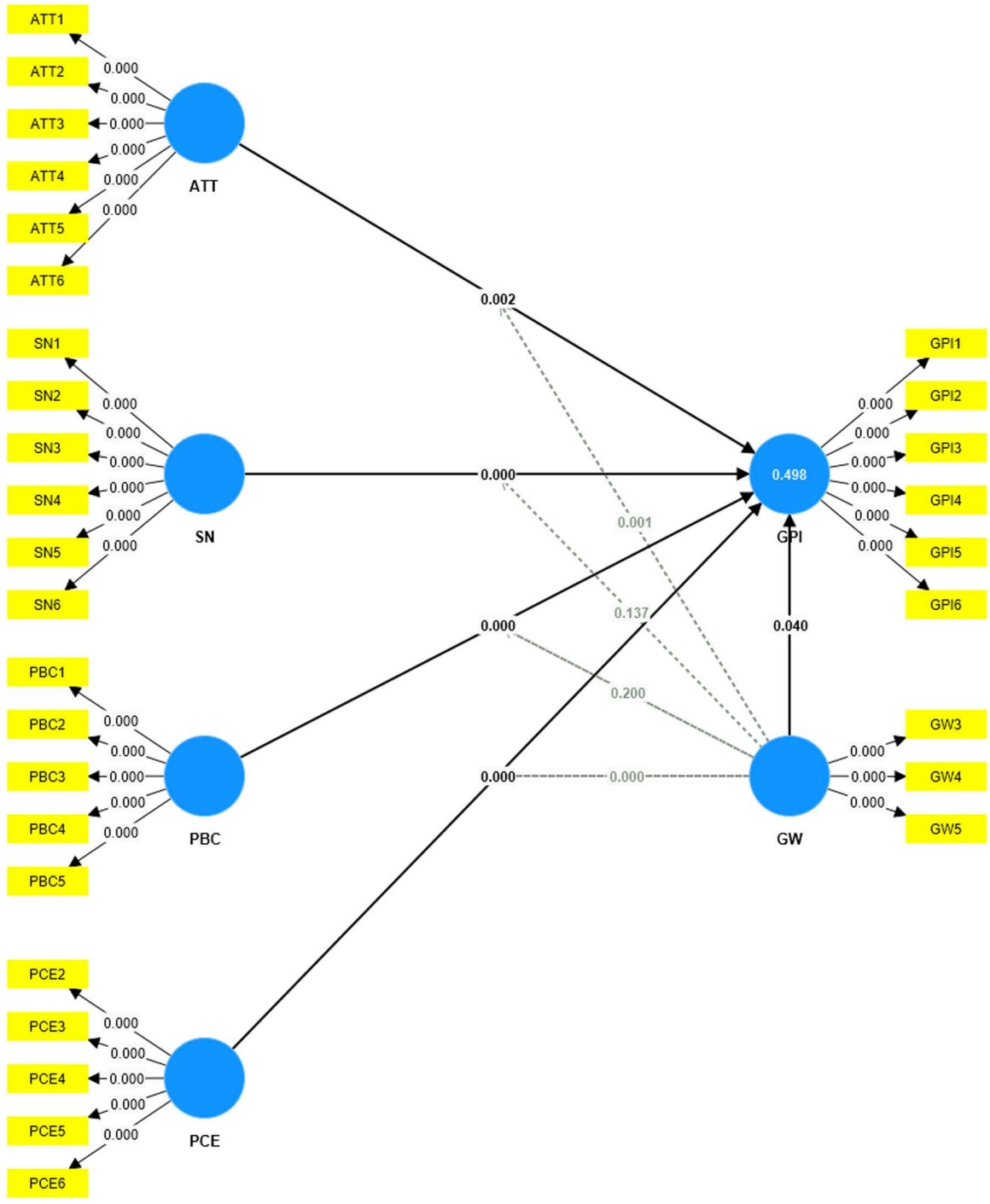

**Fig 5. Path coefficients and hypotheses testing.** It suggest that the GW*ATT interaction terms and the GW*PCE interaction term were significant.

that greenwash concerns can impact consumers' purchasing decisions. However, we extend their work by demonstrating that greenwash perception, rather than just greenwash concern, plays a crucial moderating role. The study demonstrates how greenwash perception can alter the impact of other factors on green buy intention, providing a comprehensive view

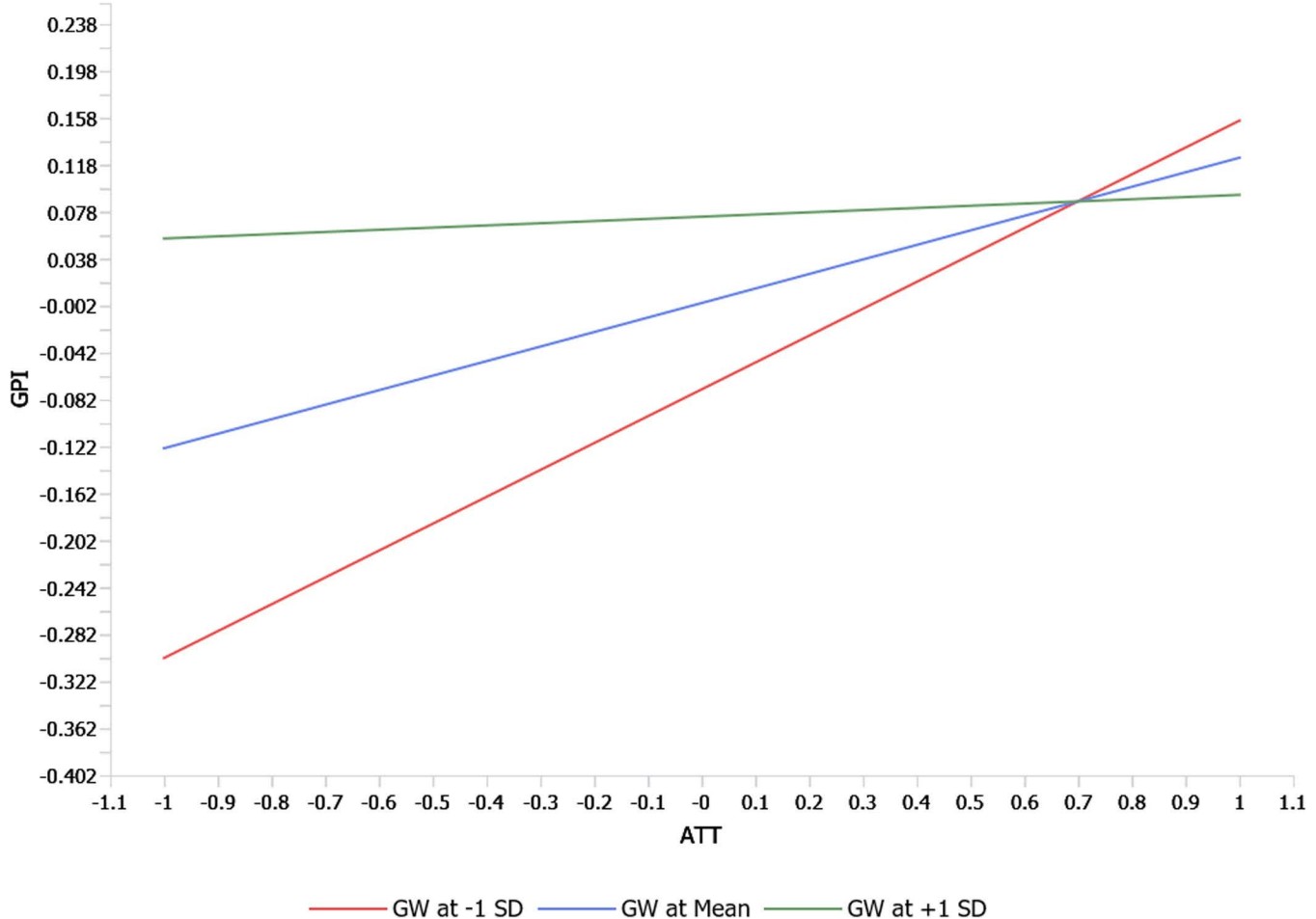

**Fig 6. Simple slope analysis GW*ATT.**

of the psychological mechanisms behind sustainable purchasing behavior, particularly in the context of fashion – a sector frequently criticized for greenwashing practices.

Furthermore, there remains a limited number of studies investigating the moderating effect of greenwash perception on various relationships. Our research has filled this critical gap by examining how greenwash perception influences the consumption of Generation Z individuals in Vietnam within the fashion industry. By doing so, we have provided unique insights into a demographic and market that have been underexplored in existing literature.

### 5.3. Managerial implications

The implications drawn from the research have significant practical relevance for various stakeholders in the context of Greenwashing among Generation Z in Vietnam's fashion industry.

For marketers and organizations, the study emphasizes the necessity of designing marketing strategies that align with Gen Z's values and concerns. As Gen Z prioritizes sustainable and eco-friendly clothing, businesses may capitalize on this demographic's desire to make a good effect by promoting the environmental benefits of their products and services. Businesses should make investments in programs that advance their environmental awareness and show a sincere dedication to sustainability. This involves training personnel, establishing sustainable practices, and clearly explaining green initiatives.

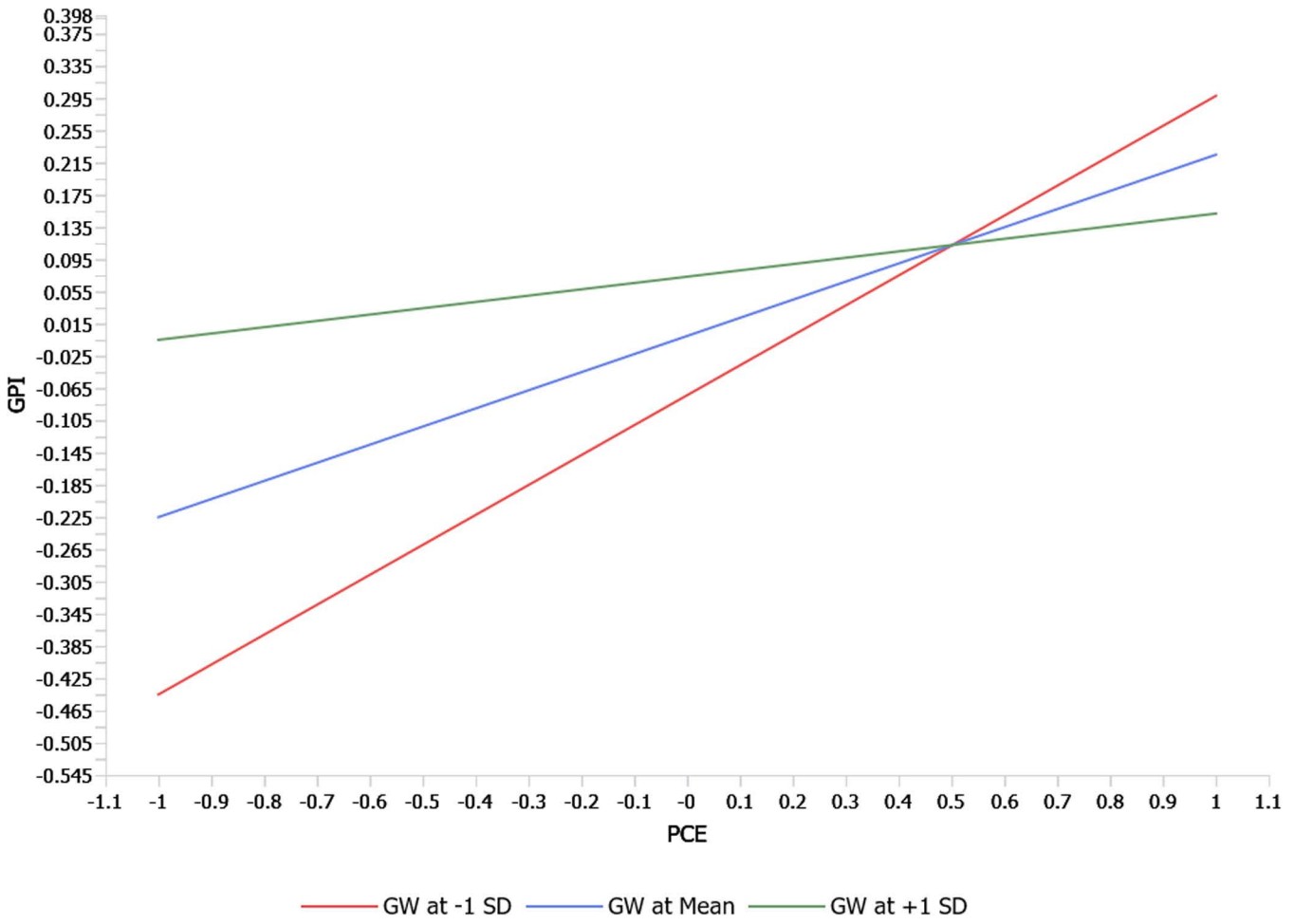

**Fig 7. Simple slope analysis GW*PCE.**

Furthermore, the findings underscored the value of openness and authenticity in green marketing. By transparently communicating the actual environmental impact of their products, businesses can strengthen consumers' belief in the positive effects of their green purchases. Customers from Generation Z penalize companies that are perceived to be inauthentic, underlining the importance of transparent communication regarding actual sustainability measures. This includes precise information on materials, manufacturing methods, and environmental impacts, which may be accompanied by relevant certifications.

The findings can also direct policymakers in encouraging sustainable consumerism and giving young people the tools they need to make wise decisions. Policymakers should impose stronger restrictions and punishments for misleading statements to level the playing field for companies and guarantee that customers have access to correct information. By enacting laws that address problems like resource usage, waste management, and labor standards, policymakers may also aid in the growth of a more sustainable fashion sector. By creating a supportive environment for sustainable businesses, policymakers can encourage innovation and drive positive change.

## 6. Limitations and future research direction

Although this study provides important new information about how Vietnam's Generation Z is affected by their perception of greenwashing, it also highlights its shortcomings and makes recommendations for future research areas [95].

Firstly, the sample size may affect the generalizability of the findingsas it does not fully capture the Vietnam's Generation Z population. Future research could broaden the scope of studies to include a more representative range of individuals from broader demographic and geographic segments for more comprehensive results. Secondly, the study is limited to the fashion industry, which does not comprehensively reflect the influence of greenwashing across other sectors. Future research could broaden its scope to encompass diverse sectors, such as electronics, food, energy, and beyond, across multiple countries. Finally, the conceptual model used a limited number of antecedents affecting green purchase intention. Future research could incorporate elements like perceived risk, green skepticism, and brand trust while examining mediating and moderating effects of consumer values and cultural influences.

## Supporting information

**S1 File. Data.**
(CSV)

## Author contributions

**Conceptualization:** Anh Duc Do.

**Data curation:** Diem Quynh Dang.

**Formal analysis:** Diem Quynh Dang.

**Investigation:** Thi Mai Bui.

**Methodology:** Anh Duc Do.

**Project administration:** Thi Mai Bui.

**Resources:** Minh Ngoc Tran.

**Software:** Minh Ngoc Tran.

**Supervision:** Dieu Linh Ha.

**Validation:** Thi Thuy Linh Phan.

**Visualization:** Thi Thuy Linh Phan.

**Writing – original draft:** Anh Duc Do.

**Writing – review & editing:** Dieu Linh Ha, Tran Bao Ngoc Le.

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
