## [Decision Letter · Decision Letter 0]

29 May 2024

PONE-D-24-05489ANTECEDENTS OF GEN Z’S GREEN PURCHASE INTENTION IN VIETNAM’S FASHION INDUSTRY WITH THE MODERATING ROLE OF GREENWASH PERCEPTIONPLOS ONE

Dear Dr. Ha,

Thank you for submitting your manuscript to PLOS ONE. After careful consideration, we feel that it has merit but does not fully meet PLOS ONE’s publication criteria as it currently stands. Therefore, we invite you to submit a revised version of the manuscript that addresses the points raised during the review process.

**ACADEMIC EDITOR**The manuscript needs to be meticulously and carefully edited. It should provide a more comprehensive review of the literature and updated information. The methods of analysis and techniques used need to be further clarified. It is particularly important for the article needs to clearly highlight the newly discovered points from the research. The format and language used are also important areas of concern for the article

I am also enclosing the comprehensive comments from 2 reviewers here for you to make necessary improvements to the article 

We look forward to receiving your revised manuscript.

Kind regards,

Thang Quyet Nguyen, Ph.D

Academic Editor

PLOS ONE

Journal Requirements:

Reviewers' comments:

Reviewer's Responses to Questions

**Comments to the Author**

1. Is the manuscript technically sound, and do the data support the conclusions?

Reviewer #1: Yes

Reviewer #2: Yes

2. Has the statistical analysis been performed appropriately and rigorously? 

Reviewer #1: Yes

Reviewer #2: Yes

3. Have the authors made all data underlying the findings in their manuscript fully available?

Reviewer #1: Yes

Reviewer #2: Yes

4. Is the manuscript presented in an intelligible fashion and written in standard English?

Reviewer #1: Yes

Reviewer #2: No

5. Review Comments to the Author

Reviewer #1: This paper is not a novelty.

The abstract is full content.

This study shows the impact of 4 factors (Subjective Norms, Attitudes, Perceived Behavioural Control, and Perceived Consumer Effectiveness) on the Green Purchase Intention of Gen Z is also not new. The citation of the paper has not been updated.

The TPB of Ajzen (1985, 1991) needs to be applied and updated new references.

In Methodology: Research Measures need to show detailed.

Data analysis should show Outer loading, Cronbach Alpha, CR, AVE …

Spending on green products monthly exchanges USD.

Some table should also be designed better.

Implications: They are not written to focus on the contents of the paper. It needs to be expanded.

Reviewer #2: Manuscript Number: PONE-D-24-05489

Title: Antecedents of gen z’s green purchase intention in Vietnam’s fashion industry with the moderating role of greenwash perception

1. Introduction:

The authors have described the current situation in the Vietnamese context. However, green purchase intention is not a new topic. The authors have failed to examine the literature and the research gap has been poorly emphasized in the introduction. One paragraph to include the academic/ theoretical background and explain why authors want to focus on this topic is necessary. Furthermore, authors might consider including the research objectives at the end of the introduction.

2. Literature review

Again, since green purchase intention and TPB have been excessively examined, authors should focus more on how to apply the model in the fashion industry for GenZ consumers. What has been explained in the literature review is too general. Furthermore, it is better to rewrite the part of the moderating hypothesis. This hypothesis development part seems to explain the direct effects only.

3. Methodology

Authors should include a few sentences to describe how they approach the participants to collect data.

93% is not the response rate. It is the completion rate.

4. Conclusions and implications

This section should be divided into 3 parts: conclusion, practical implications, and limitations and future research. The conclusion part should discuss whether the research objectives have been completed or not and mention the contribution of this study.

The authors have written quite lengthy paragraphs for the implications. However, these implications are too general. Please rewrite the parts.

5. Language and formatting

The language quality is not at the academic level. there are many grammar mistakes. Please consider using a proofreading service.

6. PLOS authors have the option to publish the peer review history of their article (what does this mean? ). If published, this will include your full peer review and any attached files.

**Do you want your identity to be public for this peer review?** For information about this choice, including consent withdrawal, please see our Privacy Policy .

Reviewer #1: No

Reviewer #2: No

---

## [Author Response · Author response to Decision Letter 1]

4 Jun 2024

RESPONSE TO REVIEWER

Title: Antecedents of gen z’s green purchase intention in Vietnam’s fashion industry with the moderating role of greenwash perception

1. Introduction:

The authors have described the current situation in the Vietnamese context. However, green purchase intention is not a new topic. The authors have failed to examine the literature and the research gap has been poorly emphasized in the introduction. One paragraph to include the academic/ theoretical background and explain why authors want to focus on this topic is necessary. Furthermore, authors might consider including the research objectives at the end of the introduction.

Author response: Thank you for the feedback. We understand that the theoretical background and research gap needs to be emphasized to highlight that the study is necessary, and the objectives of the study should be done. In response to the feedback, we have added a paragraph outline the theoretical background and the important of this topic in the manuscript. This adjustment will contribute to a more comprehensive introduction, creating the conditions for a more informative reading experience.

Change in manuscript:

1. Introduction

The majority of studies investigating green purchasing intention (GPI) have adopted a general approach, with limited research specifically focusing on the fashion industry (Casalegno et al., 2022; Yadav & Pathak, 2016), or conducted in developed countries such as China (Lu et al., 2022) and Germany (Rausch & Kopplin, 2021). To address these gaps, this study delves into GPI within the fashion industry in Vietnam, a developing country. To address these gaps, this study delves into green purchase intention within the fashion industry in Vietnam, a developing nation. Furthermore, previous studies have proposed different models of Greenwashing (Szabo & Webster, 2020; Nguyen et al., 2019), often treating Greenwash Perception as an independent variable. However, calls have been made to investigate the moderating role of Greenwash Perception to see the relationship between the variables in the extended TPB and Green Purchase Intention, this study will close a gap in previous research by considering Greenwash Perception as a moderator. Recognizing the limitations of prior research, our study focuses on the influence of greenwash on the green purchase intention of consumers, particularly Gen Z, a key target group for fashion brands.

This study examined the impact of 4 factors (Attitudes, Subjective Norms, Perceived Behavioral Control, and Perceived Consumer Effectiveness) on the Green Purchase Intention of Gen Z when the greenwashing perception is a moderating variable, applying the TPB and analyzing data through the PLS-SEM method. The following section of the study will include the literature review and hypotheses development in Section 2. The methodology will be explained in Section 3, and after that, the data analysis techniques and results will be shown in Section 4. The last part provided the implications, conclusions, and study limitations.

To achieve the research objectives, the research will answer the following questions:

(1) What factors influence Gen Z's intention to consume green fashion products in Vietnam? What is the level of impact of these factors on Gen Z's intention to consume green fashion in Vietnam?

(2) Does the perception of “greenwashing” moderate the relationship between factors and green fashion consumption intention? What is the direction and level of moderating of "greenwashing"?

2. Literature review:

Again, since green purchase intention and TPB have been excessively examined, authors should focus more on how to apply the model in the fashion industry for GenZ consumers. What has been explained in the literature review is too general. Furthermore, it is better to rewrite the part of the moderating hypothesis. This hypothesis development part seems to explain the direct effects only.

Author response: Thank you for the comment and suggestion. It is true that the Theory of Planned Behavior (TPB) has been widely discussed in various studies. Therefore, we have added more details on how previous studies have applied this model to research on fashion for Gen Z consumers. Additionally, we have focused more on explaining the moderating role of Greenwash Perception to enhance the understanding of readers and ensure consistency throughout the study.

Change in manuscript:

2. Literature review and hypothesis development

In the context of fashion and sustainable fashion, TPB has proven to be a useful model for forecasting consumer intentions, particularly Gen Z consumers. Phau et al. (2015) used TPB analysis to assess the reasons behind purchasing luxury fashion clothes manufactured in sweatshops. They looked at two scenarios: the desire to pay more for fashion apparel that is not sweatshop-made and the desire to avoid purchasing luxury brands that are. Rausch et al. (2020) endeavored to verify TPB in the context of green product consumption by utilizing the expanded version of TPB, which encompasses environmental concerns. The authors discovered that while customers' concerns about greenwashing have a detrimental impact on this relationship, attitude toward sustainable clothes influences purchase intention most. A model based on the extension of TPB was developed in Ge's (2024) research to determine the elements of influencer marketing that impact the primary reasons why Gen Z consumers intend to buy sustainable fashion products. These factors include attitudes toward sustainable fashion influencers, friends' opinions influenced by influencers, and the affordable products that influencers recommend.

2.5 Moderating role of Greenwash Perception

Green marketing has developed in response to consumers' increased desire for products and services that have a positive environmental impact as they become more concerned about environmental protection in the context of globalization (Mishra & Sharma, 2010). Using green marketing methods has become standard practice for companies looking to get a competitive edge and draw in eco-aware customers. It is imperative to acknowledge that not all assertions made via green marketing precisely mirror an organization's true environmental practices, which may result in "greenwashing" (Szabo & Webster, 2021). The phrase "greenwashing" refers to the false and dishonest representations made by certain companies regarding the environmental friendliness of their products or services (Parguel et al., 2011). It is important to recognize greenwashing because, in the absence of it, well-meaning consumers may be duped into believing that the choices they make would contribute to environmental preservation. The term "greenwashing" actually has a lot of definitions, especially in the last several years. Certain studies have resulted in advancements and improved precision in comprehending the notion of greenwashing. Walker et al. (2012) define greenwashing as the distinction made by a business between "substantial" and "symbolic" social actions; the former narrows the focus from environmental issues to the latter through the use of image advertising, confusing terminology, and imagery.

While greenwashing can theoretically help businesses boost earnings and improve their standing and reputation, it has a detrimental effect on the environment and sustainable social behaviors. Numerous studies have shown the detrimental effects of greenwashing, as the trust that stakeholders have in green brands has been significantly damaged by corporations' attempts to utilize greenwashing in product promotion (Guo et al., 2017). In addition to having a detrimental effect on customer behavioral intentions, greenwashing also has a negative influence on business social responsibility and reputation (Cordero et al., 2021).

Recent years have seen a significant development in the reality of "greenwashing" in the fashion business, particularly as people's awareness of environmental preservation and sustainable living has grown. Studies have been done to look at the reasons behind greenwashing and how it impacts businesses, investors, and other stakeholders. Customers' capacity to distinguish between green advertising messaging and a company's true environmental commitment is known as their perception of greenwashing (Nyilasy, 2014). Put differently, customers' psychological assessments regarding a company's potential to misrepresent its environmental impact and hide its true environmental message are known as perceptions of greenwashing.

Limited research exists on how corporate greenwashing, within the TPB framework, influences consumer green purchase intentions. This study proposes a theoretical model where Greenwash Perception moderates the relationships between the core extended TPB constructs and Green Purchase Intention.

Consumers with positive attitudes towards green products are more likely to exhibit green purchase behavior (Suki, 2013). However, greenwash perception can moderate this relationship. If consumers perceive greenwashing, their positive attitude towards green products may weaken, leading to a decreased likelihood of green purchase.

Subjective norms represent societal influences on consumer behavior (De Pelsmacker et al., 2006). Consumers are affected by their social circles and the perceived social acceptability of various activities. Greenwash perception might help to attenuate this association. If consumers learn of a company's greenwashing, they may be less likely to buy the product because of fear of social repercussions for supporting ecologically hazardous actions.

Perceived behavioral control relates to customers' belief in their capacity to do a given activity (De Pelsmacker, 2006). In the context of green purchase intention, this translates to a consumer's confidence in their ability to make green choices. Studies have shown a positive correlation, indicating that consumers who feel empowered to make a difference through their purchases are more likely to do so. However, when consumers encounter greenwashing tactics, it can erode their trust in companies' environmental claims. This skepticism can lead them to question the effectiveness of their individual choices, hence, discouraging consumers from engaging in green purchasing despite their perceived ability to do so.

Similar to Perceived Behavioral Control, Perceived Consumer Effectiveness plays a role in influencing purchasing intention, refers to a consumer's belief in the impact of their actions on achieving a desired outcome. In the context of green practices, this translates to a consumer's belief that their green purchases can make a difference for the environment. However, Greenwash Perception may undermine this relationship. Exposed to greenwashing, consumers may doubt the true environmental benefit of green products, leading them to question the effectiveness of their choices despite a strong belief in making a difference.

3. Methodology:

Authors should include a few sentences to describe how they approach the participants to collect data.

93% is not the response rate. It is the completion rate.

Author response: Thank you for the comment. We have reviewed the manuscript and made the necessary adjustments.

Change in manuscript:

3.4 Participants

The authors collected data through an online survey using Google Forms and distributed it on social media platforms. Out of the 500 responses that were recorded, 467 proved to be valid after removing the invalid ones. With a 93% completion rate, this shows a high level of participant engagement. These comprise 28.9% male and 71.1% female. Of the Gen Z participants surveyed, 76.9% are 19 - 22 years old, 17.3% are between the ages of 15 and 18, and 5.8% are between the ages of 23 and 26. The descriptive statistics of the survey are shown in the table 1 below.

4. Conclusions and implications:

This section should be divided into 3 parts: conclusion, practical implications, and limitations and future research. The conclusion part should discuss whether the research objectives have been completed or not and mention the contribution of this study.

The authors have written quite lengthy paragraphs for the implications. However, these implications are too general. Please rewrite the parts.

Author response: Thank you for the comment and suggestions. Based on the comments, we have divided into 3 parts: Conclusions, implications, and limitations and future research direction. Moreover, we also made some adjustments to the implications part to make it more specific and focused.

Change in manuscript:

5. Conclusions and Implications

5.1. Conclusions

This study proposes an extensive model to show how Gen Z customers in Vietnam's fashion sector intend to make green purchases. According to the study's model, the antecedents of intention to buy green products include attitude, subjective norms, perceived behavior control, and perceived consumer effectiveness. Furthermore, by adding greenwash perception as a moderator variable, the study unveils the nuanced influence of greenwash perception as a moderator, significantly impacting the relationships between key determinants such as attitudes and perceived consumer effectiveness. The negative influence of greenwash perception shows an increase in the awareness of Generation Z about the sustainability of products but also sensitivity to insincere green promises.

The study's findings reveal a robust relationship between various factors and the purchase intentions of young consumers, particularly in the context of green products, with a specific focus on green clothing. The predictive elements encompass attitudes and subjective norms, perceived behavioral control. The predictive elements encompass Attitudes and Subjective Norms, Perceived Behavioral Control, and Perceived Consumer Effectiveness. This outcome is consistent with existing research across diverse domains, including studies on green vegetable purchasing in developing countries (Nguyen et al., 2019), the acquisition of organic food by young consumers aged 18-30 in China (Ahmed et al., 2021), and the purchase intentions of Chinese consumers towards green products (Liu et al., 2020).

The alignment of our results with these varied research efforts underscores the applicability and relevance of the theory of planned behavior to our study. The empirical evidence supports the contention that attitudes, subjective norms, and perceived behavioral control exert positive effects on the purchase intentions of young consumers, reinforcing the suitability of the chosen theoretical framework and its associated measures for our investigation.

Some studies assert a direct influence of greenwashing on the intention to make environmentally friendly purchases, as evidenced by Zhang et al. (2018). Simultaneously, there is a body of research exploring the role of greenwashing as a moderator (Zaidi et al., 2019; Bulut et al., 2021; Wu & Liu, 2022). This present study delves into the moderating function of perceived greenwash. In our ongoing study, we are specifically exploring the moderating role of perceived greenwash. Greenwashing perceptions indeed appear to influence consumer decisions on the intention-formation level as they were found to moderate the relation between attitude and green purchase intention according to Rausch & Kopplin (2021), towards sustainable clothes and purchase intention for sustainable clothes, the same result is also proved in our study. Moreover, Rausch & Kopplin's study (2021) yielded similar results to our investigation, suggesting insufficient evidence to demonstrate that perceived greenwash moderates the relationship between subjective norms and green purchase intentions. A research by Rejikumar (2016) has also demonstrated the same negative moderator role of Perceived Greenwash between Perceived Consumer Effectiveness and Green Purchase Intention.

5.2. Theoretical implications

The research has developed a model to study the factors influencing green purchase intention, with the addition of the moderating effect of greenwashing perception on these relationships with a focus on Generation Z in Vietnam's fashion sector. This contributes to the theoretical diversity of topics related to green consum

---

## [Decision Letter · Decision Letter 1]

9 Jan 2025

PONE-D-24-05489R1ANTECEDENTS OF GEN Z’S GREEN PURCHASE INTENTION IN VIETNAM’S FASHION INDUSTRY WITH THE MODERATING ROLE OF GREENWASH PERCEPTIONPLOS ONE

Dear Dr. Ha,

Thank you for submitting your manuscript to PLOS ONE. After careful consideration, we feel that it has merit but does not fully meet PLOS ONE’s publication criteria as it currently stands. Therefore, we invite you to submit a revised version of the manuscript that addresses the points raised during the review process.

We look forward to receiving your revised manuscript.

Kind regards,

Thang Quyet Nguyen, Ph.D

Academic Editor

PLOS ONE

Journal Requirements:

Additional Editor Comments:

The manuscript needs to be meticulously and carefully edited. It should provide a more comprehensive review of the literature and updated information. The methods of analysis and techniques used need to be further clarified. It is particularly important for the article needs to clearly highlight the newly discovered points from the research. The format and language used are also important areas of concern for the article.

I am also enclosing the comprehensive comments from reviewers here for you to make necessary improvements to the article

Reviewers' comments:

Reviewer's Responses to Questions

**Comments to the Author**

1. If the authors have adequately addressed your comments raised in a previous round of review and you feel that this manuscript is now acceptable for publication, you may indicate that here to bypass the “Comments to the Author” section, enter your conflict of interest statement in the “Confidential to Editor” section, and submit your "Accept" recommendation.

Reviewer #3: All comments have been addressed

Reviewer #4: All comments have been addressed

2. Is the manuscript technically sound, and do the data support the conclusions?

Reviewer #3: Yes

Reviewer #4: Yes

3. Has the statistical analysis been performed appropriately and rigorously? 

Reviewer #3: Yes

Reviewer #4: Yes

4. Have the authors made all data underlying the findings in their manuscript fully available?

Reviewer #3: Yes

Reviewer #4: Yes

5. Is the manuscript presented in an intelligible fashion and written in standard English?

Reviewer #3: Yes

Reviewer #4: Yes

6. Review Comments to the Author

Reviewer #3: 1. The abstract showed a relevant study in the paper, clear objective, key findings and results. There are some following points that should rewrite for improvement:

- The methodology should describe more information such as number of respondents, how to collect data, how to process the collected data,...

- The implications should be clearly.

2. The introduction presented the background of the study, clearly gaps, theories and research questions. The introduction should consider the citations and explain how to apply TBP and PLS-SEM in this study.

3. The literature review showed the TPB and developed 8 hypotheses in which greenwash perception is the moderators. The review should streamline the structure with subheading consisting the concepts, theories, previous research and hypotheses development. The term of greenwash perception, purchase intention should mention in the concept and more clearly. The recent research should argument more by evidence citations.

4. The methodology showed the research measures, data collection, data analysis, participants. The methodology should explain more how to build the scales and data processing. The content of participants should mention in research results.

5. The results showed concisely reliability and value. The results should arrange the structures of subheading that are suitable for data processing. The results should explain the finding of the study.

6. The conclusion, implications and limitations are relevance with the study. However the implications should based on the observed variables to suggest more details.

7. Please check carefully the formatting and typos.

Reviewer #4: Based on the research content, I would like to provide the following feedback to the author:

- The study has chosen a topic that aligns with the current trend of sustainable development. Investigating the impact of "greenwashing" on Gen Z's green purchasing intention is a novel and highly practical direction, especially in the context of the Vietnamese fashion industry, which is undergoing a transition to a sustainable business model.

- The study used appropriate methodologies and conducted a rigorous data analysis. The use of PLS-SEM and simple slope analysis has enabled the author to accurately and reliably test the research hypotheses.

- The research findings provide useful information for fashion businesses in developing effective business and contribute to raising consumer awareness of "greenwashing".

Some suggestions for this study:

- The context and rationale for choosing Vietnam as the research setting need to be clarified. The author should provide more information about the sustainable fashion market in Vietnam, the challenges and opportunities for businesses and consumers.

- The author should discuss the limitations of the study in more depth and propose more specific directions for future research.

Overall, this is a promising study that has the potential to contribute to the development of consumer behavior research. After being revised and supplemented based on the above suggestions, the paper will have higher scientific and practical value.

7. PLOS authors have the option to publish the peer review history of their article (what does this mean? ). If published, this will include your full peer review and any attached files.

**Do you want your identity to be public for this peer review?** For information about this choice, including consent withdrawal, please see our Privacy Policy .

Reviewer #3: **Yes: ** Trung Bao

Reviewer #4: No

---

## [Author Response · Author response to Decision Letter 2]

12 Feb 2025

RESPONSE TO REVIEWER

Title: Antecedents of gen z’s green purchase intention in Vietnam’s fashion industry with the moderating role of greenwash perception

Reviewer #3:

1. The abstract showed a relevant study in the paper, clear objective, key findings and results. There are some following points that should rewrite for improvement:

- The methodology should describe more information such as number of respondents, how to collect data, how to process the collected data,...

Authors’ response: Thank you for your feedback on the methodology section. We have now included details on the number of respondents, data collection process, and data analysis methods to ensure clarity.

Change in manuscript:

Abstract

While buying sustainable fashion items is becoming more and more popular worldwide, the concept of green fashion is still relatively new in Vietnam. Besides, many fashion brands use “greenwashing” to deceive consumers and encourage their purchases. This study aims to examine the drivers of green purchasing intention among Gen Z in Vietnam’s fashion industry with greenwash perception as the moderating factor. The conceptual model was analyzed using structural equation modeling with the bootstrapping method based on data gathered from 467 Vietnamese Gen Z customers.

- The implications should be clearly.

Authors’ response: Thank you for your recommendation, the authors have improved the abstract as follows:

Change in manuscript:

Abstract

The findings reveal significant positive influences of attitude, subjective norms, perceived behavioral control, and perceived consumer effectiveness on green purchase intention. Greenwash perception serves as a significant negative moderator, enhancing the relationship between the determinants and green purchasing intention. The results hold significant implications for businesses, encouraging them to embrace transparent and genuine sustainability practices instead of engaging in greenwashing. This can be achieved by clearly communicating their initiatives, offering third-party certifications, and conducting educational campaigns. The study provides original contributions to the existing body of literature and offers recommendations for future research in the context of developing countries.

2. The introduction presented the background of the study, clearly gaps, theories and research questions. The introduction should consider the citations and explain how to apply TBP and PLS-SEM in this study.

Authors’ response: Thank you for your valuable feedback on the introduction. We have carefully reviewed the comment and made the following improvements. First, ưe have included relevant citations to strengthen the background and theoretical foundation of the study. Specifically, we have cited recent key studies that highlight the gaps in the existing literature and justify the need for this research. Second, we understand that it is important to explain the role of TPB and the use of PLS SEM as our primary analytical method, so that we have made several changes to our introductory.

Change in manuscript:

Introduction

Green consumption has become a global trend, originating in developed countries and gradually spreading to developing nations like Vietnam (Thi Tuyet Mai, 2019). To promote environmental conservation, the United Nations introduced the Sustainable Development Goals (SDGs), with Goal 12 emphasizing sustainable production and consumption to reduce the ecological footprint. While researchers have extensively studied green consumption in various fields (Bryła, 2019; Ahmed et al., 2020), there is still limited research on sustainable fashion consumption, particularly in Asia's emerging markets. Consequently, to broaden the body of current research, this study focuses on the Vietnamese fashion sector, where the idea of "green fashion" was still relatively new.

In Vietnam, the government has implemented the National Strategy for Green Growth for 2021–2030 with a vision for 2050. The strategy includes initiatives to encourage sustainable consumption and promote greener lifestyles. Alongside rapid economic growth and a booming fashion sector (Nayak, 2021), Vietnamese consumers are increasingly aware of sustainability. However, the sustainable fashion industry in Vietnam faces significant challenges, including high production costs, limited resources (Zhang et al., 2018; Jackman & Moore, 2021), and deceptive practices like greenwashing, where companies create false impressions of environmental friendliness (Nguyen et al., 2019). Additionally, Vietnam still lacks comprehensive regulations and enforcement mechanisms to address these issues effectively. To be more precise, greenwashing refers to creating a false impression or presenting misleading information about a company's products being environmentally friendly. It arises from combining poor environmental performance with positive communication about sustainability (Nguyen et al., 2019). While greenwashing has been widely studied, there is a notable lack of research on its prevalence in the fashion industry within developing Asian countries (Yang et al., 2020). This is crucial, as consumers are often willing to pay more for products from companies genuinely committed to sustainability (Chen et al., 2018).

The research focuses on Generation Z (born between 1997 and 2012) for their strong awareness of ethical and environmental issues. This generation is often willing to pay more for eco-friendly products and has a significant influence on market trends (Casalegno et al., 2022; Anh Do et al., 2023). However, despite their environmentally conscious attitudes, Gen Z remains the largest consumer group in the fast fashion industry, accounting for about 40% of global sales. By 2025, Gen Z will include 2 billion individuals worldwide and will make up approximately 25% of Vietnam’s labor force, significantly influencing household and market consumption decisions (Do et al., 2023). Given their dual role as eco-conscious consumers and primary contributors to fast fashion consumption, it is critical to examine Gen Z's green purchase intentions in Vietnam’s fashion sector.

Therefore, most studies on green purchasing intention have taken a general approach, with limited focus on the fashion industry (Casalegno et al., 2022; Yadav & Pathak, 2016) or specific populations like Gen Z. Furthermore, while greenwashing has been studied as an independent variable (Nguyen et al., 2019; Szabo & Webster, 2020), there is a lack of research exploring its moderating role in shaping consumer behavior. To address these gaps, this study focuses on the Vietnamese fashion industry and investigates the influence of greenwashing perception on Gen Z's intention to purchase green fashion products.

To achieve the research concern, the research will answer the following questions:

RQ1: How do four drivers (attitudes, subjective norms, perceived behavioral control, and perceived consumer effectiveness) influence the Gen Z's green purchase intention in Vietnam’s fashion industry?

RQ2: How does the perception of greenwashing moderate the relationship between these factors and Gen Z’s intention to consume green fashion products?

This study provides both theoretical and practical contributions by addressing these questions. By extending the Theory of Planned Behavior (TPB) model with four drivers (attitudes, subjective norms, perceived behavioral control, and perceived consumer effectiveness) and underscoring the role of moderating variable of greenwashing perception serving, this study provides a thorough understanding of factors influencing green purchasing intention among Gen Z in Vietnam’s fashion industry. The research was utilized the extension of TPB and used the PLS-SEM method for data analysis. The following section of the study will include literature review and the development of hypotheses. The methodology will be explained in Section 3, and after that, the data analysis techniques and results will be shown in Section 4. The last part provided the conclusions, implications, and study limitations.

3. The literature review showed the TPB and developed 8 hypotheses in which greenwash perception is the moderators.

- The review should streamline the structure with subheading consisting the concepts, theories, previous research and hypotheses development.

Authors’ response: Thank you for your constructive feedback. We have reorganized the literature review to include clear subheadings involve: (1) Greenwashing and Greenwash Perception, (2) Green Purchase Intention, (3) Extended Theory of Planned Behavior, and (4) Hypothesis Development. This restructuring ensures a logical and coherent presentation of the key elements.

- The term of greenwash perception, purchase intention should mention in the concept and more clearly. The recent research should argument more by evidence citations.

Authors’ response: Thank you for your constructive feedback. The authors have also expanded the definitions and discussions of greenwash greenwash perception and green purchase intention. These terms are clearly explained under each subheading, highlighting their importance to the study.

Changes in manuscript:

2. Literature Review and Hypothesis Development

2.1. Greenwashing and Greenwashing Perception

Green marketing has developed in response to consumers' increased desire for products and services that have a positive environmental impact as they become more concerned about environmental protection in the context of globalization (Szabo & Webster, 2020). Using green marketing methods has become standard practice for companies looking to get a competitive edge and draw in eco-aware customers. It is imperative to acknowledge that not all assertions made via green marketing precisely mirror an organization's true environmental practices, which may result in "greenwashing" (Szabo & Webster, 2021). The phrase "greenwashing" refers to the false and dishonest representations made by certain companies regarding the environmental friendliness of their products or services (Parguel et al., 2011). It is important to recognize greenwashing because, in the absence of it, well-meaning consumers may be duped into believing that their choices would contribute to environmental preservation. The term "greenwashing" has a lot of definitions, especially in the last several years. The practice of misleadingly disclosing "green" activities is known as "greenwashing" (Lee & Raschke, 2023; Seele & Schultz, 2022). It may be dependent on external factors, incentives, or pressures that define the institutional context in which it takes place (Zharfpeykan, 2021; Velte, 2022; Seele & Schultz, 2022; Li et al., 2022). Recent years have seen a significant development in the reality of "greenwashing" in the fashion business, particularly as people's awareness of environmental preservation and sustainable living has grown (Munir & Mohan, 2022; Alizadeh et al., 2024).

A negative perception component, greenwash perception is defined as the public or consumers recognizing and perceiving inaccurate or deceptive comments made by businesses about their environmental actions (Chen & Chang, 2012). This view reflects people's misgivings about whether a corporation is carrying out its environmental protection obligations and their cynicism about the sincerity of the company's green marketing initiatives (Pomering & Johnson, 2009). Customers generally hold the view that environmentally friendly products are safer and more ecologically friendly (Pekersen & Canöz, 2022), which results in a generally favorable perception of these items (Park & Lin, 2020; Kement et al., 2023). Customers are more inclined to doubt advertising claims when they see examples of greenwashing, though, and this can have a detrimental effect on how they feel about the products (Chang & Hung, 2023). Thus, greenwash perception has emerged as a key metric for assessing consumer trust and business social responsibility initiatives in the present climate of growing environmental consciousness (Munir & Mohan, 2022).

2.2. Green Purchase Intention

Green purchase intention refers to the consumer’s willingness or readiness to purchase environmentally friendly products (Ruangkanjanases et al., 2020). When evaluating customers' green intentions, several factors are crucial, including beliefs, needs, values, motivation, knowledge, demography, and attitudes (Choi & Johnson, 2019). A person's inclination and desire to favor environmentally friendly products over non-eco-friendly ones is theorized while making a purchase decision (Spielmann, 2020). Customers identify their need for a product during the evaluation process, which influences their choice to buy (Sullivan and Kim, 2018). In the context of green purchasing, this intention manifests when consumers consciously decide to buy eco-friendly products, aligning their actions with a commitment to reducing environmental harm (Ahmed et al., 2022). Based on the TPB model, green purchase intention builds on the broader concept of general purchase intention by raising consumer awareness of environmental sustainability, health, and nature while concentrating on eco-friendly products (Ali et al., 2022). Research suggests that a positive attitude toward green products significantly enhances the likelihood of forming green purchase intention (Amoako et al., 2020). Green purchase intention varies across cultures, individuals, and genders, highlighting the contextual nature of consumer behavior (Sreen et al. 2018). Identifying and appreciating green attributes in products not only motivates consumers to make environmentally conscious decisions but also positions them as responsible citizens who contribute to broader societal changes (Hamzah & Tanwir, 2021; Wang, Zaman and Alvi, 2022).

4. The methodology showed the research measures, data collection, data analysis, participants. - The methodology should explain more how to build the scales and data processing.

Authors’ response: Thank you for your insightful feedback. Thank you for your insightful feedback. To enhance clarity, we have restructured the methodology section into three subheadings: (1) Data Collection and Sample, (2) Research Measures, and (3) Data Analysis. We have also added details on how the research scales were developed, including the process of adapting and validating items from previous studies in the Research Measure section. This section now also describes the rationale for selecting specific measurement items and their relevance to the study objectives.

Changes in manuscript:

3. Methodology

3.1. Data collection and sample

The authors collected data through a quota convenience sampling method. The survey targeted in this study were Vietnamese Generation Z customers, who were born between 1997 and 2012. Data collection was conducted anonymously through an online survey distributed via Google Forms across various media channels and forums. This approach helped maintain participant neutrality and enhance data accuracy. A total of 500 responses were recorded, of which 467 were deemed valid after excluding incomplete or inconsistent entries. The survey achieved a high completion rate of 93%, reflecting strong participant engagement.

Out of 467 appropriate respondents, there are 28.9% of male respondents compared to 71.1% female respondents. Of the Gen Z participants surveyed, 76.9% are 18 - 22 years old, 17.3% are between the ages of 13 and 17, and 5.8% are between the ages of 23 and 28. Most survey respondents reside in Northern Vietnam, with 343 individuals (73.4%), followed by Southern Vietnam with 66 respondents (14.1%), and Central Vietnam accounting for 9.4%. Regarding occupation, the largest group comprises students, totaling 434 participants (92.9%), followed by employees with 23 individuals (4.9%). Business owners, freelancers, and others make up 0.9%, 0.4%, and 0.9%, respectively. Most participants spend less than 500,000 VND per month on green products (84.2%), while a smaller proportion spend between 500,000 and 1,000,000 VND (14.1%), and only a very small percentage spends over 1,000,000 VND (1.7%). The descriptive statistics of the survey are shown in Table 1 below.

3.2. Research Measures

Multiple items on a 5-point Likert-type Scale (1 = "Strongly disagree

---

## [Decision Letter · Decision Letter 2]

4 May 2025

ANTECEDENTS OF GEN Z’S GREEN PURCHASE INTENTION IN VIETNAM’S FASHION INDUSTRY WITH THE MODERATING ROLE OF GREENWASH PERCEPTION

PONE-D-24-05489R2

Dear Dr. Ha,

We’re pleased to inform you that your manuscript has been judged scientifically suitable for publication and will be formally accepted for publication once it meets all outstanding technical requirements.

Kind regards,

Thang Quyet Nguyen, Ph.D

Academic Editor

PLOS ONE

Additional Editor Comments (optional):

Congratulations! The reviewers all agree with the revisions of your manuscript. It is accepted for publication. However, the manuscript still requires some minor revisions before it can be published, including:

1. Review and correct grammar and writing style;

2. Verify symbols and terminology;

3. Expand the conclusion to better highlight the manuscript's contributions

Reviewers' comments:

Reviewer's Responses to Questions

**Comments to the Author**

1. If the authors have adequately addressed your comments raised in a previous round of review and you feel that this manuscript is now acceptable for publication, you may indicate that here to bypass the “Comments to the Author” section, enter your conflict of interest statement in the “Confidential to Editor” section, and submit your "Accept" recommendation.

Reviewer #4: All comments have been addressed

2. Is the manuscript technically sound, and do the data support the conclusions?

Reviewer #4: Yes

3. Has the statistical analysis been performed appropriately and rigorously? 

Reviewer #4: Yes

4. Have the authors made all data underlying the findings in their manuscript fully available?

Reviewer #4: Yes

5. Is the manuscript presented in an intelligible fashion and written in standard English?

Reviewer #4: Yes

6. Review Comments to the Author

Reviewer #4: The manuscript presents a compelling contribution to the field of green purchase intention in Vietnam, with particular strengths in methodology and clarity of content. The research question is well-defined, and the methods are appropriate for the study objectives. The conclusions are supported by the data, and the manuscript is generally well-organized and clearly written.

Ethical Concerns: At this time, there are no apparent concerns regarding research ethics or publication ethics. The authors appear to have followed standard ethical procedures

No signs of duplicate publication were identified during this review. The work appears original and unpublished elsewhere, based on the information provided.

Additional Comments:

- In table 5, the variable names should be changed to GW*ATT, GW*SN, GW*PBC, and GW*PCE instead of the symbol GP to unify the symbols with other contents.

- Similarly correct the contents of figures 6, 7,

7. PLOS authors have the option to publish the peer review history of their article (what does this mean? ). If published, this will include your full peer review and any attached files.

**Do you want your identity to be public for this peer review?** For information about this choice, including consent withdrawal, please see our Privacy Policy .

Reviewer #4: No

---

## [Editor Report · Acceptance letter]

PONE-D-24-05489R2

PLOS ONE

Dear Dr. Ha,

I'm pleased to inform you that your manuscript has been deemed suitable for publication in PLOS ONE. Congratulations! Your manuscript is now being handed over to our production team.

Kind regards,

on behalf of

Professor Thang Quyet Nguyen

Academic Editor

PLOS ONE